# Affecting the Effectors: Regulation of *Legionella pneumophila* Effector Function by Metaeffectors

**DOI:** 10.3390/pathogens10020108

**Published:** 2021-01-22

**Authors:** Ashley M. Joseph, Stephanie R. Shames

**Affiliations:** Division of Biology, Kansas State University, Manhattan, KS 66506, USA; amjoseph@ksu.edu

**Keywords:** *Legionella pneumophila*, metaeffector, effector

## Abstract

Many bacterial pathogens utilize translocated virulence factors called effectors to successfully infect their host. Within the host cell, effector proteins facilitate pathogen replication through subversion of host cell targets and processes. *Legionella pneumophila* is a Gram-negative intracellular bacterial pathogen that relies on hundreds of translocated effectors to replicate within host phagocytes. Within this large arsenal of translocated effectors is a unique subset of effectors called metaeffectors, which target and regulate other effectors. At least one dozen metaeffectors are encoded by *L. pneumophila*; however, mechanisms by which they promote virulence are largely unknown. This review details current knowledge of *L pneumophila* metaeffector function, challenges associated with their identification, and potential avenues to reveal the contribution of metaeffectors to bacterial pathogenesis.

## 1. Introduction

Bacterial pathogens use a myriad of virulence strategies to parasitize eukaryotic hosts. A well-established virulence strategy is use of macromolecular secretion systems to translocate bacterial protein virulence factors, termed effector proteins, directly into infected host cells [1]. *Legionella pneumophila* is a natural intracellular pathogen of freshwater amoebae and the etiological agent of Legionnaires’ Disease, a severe inflammatory pneumonia resulting from bacterial replication within alveolar macrophages. To replicate intracellularly, *L. pneumophila* employs a type IVB secretion system called Dot/Icm to translocate a massive arsenal of over 300 individual effector proteins into the host [2,3]. Collectively, *L. pneumophila* effectors facilitate biogenesis of the *Legionella*-containing vacuole (LCV), an endoplasmic reticulum-derived compartment that evades lysosomal fusion and serves as *L. pneumophila*’s intracellular replicative niche. The status quo pertaining to bacterial effectors is that they specifically target host proteins and pathways. However, *L. pneumophila* encodes a family of effectors, termed metaeffectors, which function as “effectors of effectors” through targeting and regulating the function of other effectors. Metaeffectors contribute to *L. pneumophila* virulence and provide an additional mechanism by which bacteria regulate effector functions within host cells. Here, we discuss current knowledge pertaining to *L. pneumophila* metaeffectors and conclude with the importance of future investigation into these important virulence factors within both the *Legionella* genus and other bacterial pathogens.

## 2. Identification and Function of *L. pneumophila* Metaeffectors

The term “metaeffector” was coined a decade ago when Kubori and colleagues discovered that the effector LubX spatiotemporally regulates the effector SidH within *L. pneumophila*-infected host cells [4]. LubX contains two regions with similarity to eukaryotic U-box domains, and functions as E3 ubiquitin ligase within eukaryotic cells [5]. In conjunction with UbcH5a or UbcH5c E2 enzymes, LubX polyubiquitinates the host kinase, Clk1 [4,6]. However, LubX additionally co-opts E2 enzymes to ubiquitinylate its cognate effector, SidH, leading to its proteasomal degradation (Figure 1) [4]. Like the majority of *L. pneumophila* effectors, genetic deletion of *sidH* has no discernable effect on intracellular replication within macrophages, and the function of SidH within host cells has yet to be elucidated [4,6]. However, SidH is a paralog of the *L. pneumophila* effector SdhA, which promotes *L. pneumophila* intracellular replication through maintenance of LCV integrity [4,7,8]. Thus, SidH may contribute to maintaining the integrity of the LCV during early infection. In a *Drosophila melanogaster* infection model, *L. pneumophila* ∆*lubX* mutants are hyper-lethal. However, loss of *lubX* results in decreased bacterial burden in flies compared to wild-type, ∆*sidH* and ∆*sidH*∆*lubX L. pneumophila* strains [4]. However, loss of *lubX* has no discernable effect on *L. pneumophila* replication within mouse bone marrow-derived macrophages, suggesting that SidH may be detrimental in the absence of LubX specifically in vivo. It would be valuable to reveal whether loss of LubX-mediated regulation of SidH is also deleterious to *L. pneumophila* replication in a mouse model of Legionnaires’ Disease. Interestingly, LubX expression peaks when the cells are nearing the stationary phase; much later than the critical window for SidH degradation, suggesting that mediation of SidH toxicity may not be the apogee of LubX activity [4].

Temporal regulation of effector translocation is likely important for other effector–metaeffector pairs. The metaeffector SidJ regulates the SidE family of effectors (SidE/SdeABC) to facilitate biogenesis of the LCV (Figure 1) [9,10]. SidJ is one of very few effectors individually important for *L. pneumophila* intracellular replication [11]. While expression and translocation of the SidE effectors peaks during early infection, SidJ translocation increases gradually over the course of infection [10,11]. The SidE effectors are mono-ADP-ribosyltransferases that ligate ubiquitin to Rab GTPases independently of E1 and E2 enzymes [12,13,14]. SidJ is a calmodulin-dependent glutamylase that spatiotemporally regulates the SidE effectors by breaking phosphodiester bonds between ubiquitin- and SidE-modified substrates [14]. SidJ is a calmodulin-dependent glutamylase that temporally regulates the function of the SidE effectors [14,15,16,17]. SidJ polyglutamylates Glu860 of the SidE family effector SdeA, leading to its inactivation. In the absence of SidJ, SdeA fails to depart from the LCV surface, but robustly ubiquitinates several Rab and Rag GTPases (Figure 1) [10,14]. While SidE effectors are important at early stages of infection, their prolonged activity is deleterious to *L. pneumophila*. Delayed translocation of SidJ relative to the SidE family enables precise temporal regulation of SidE effector function [10]. Timing of SidJ translocation is facilitated by an internal secretion signal, present in addition to its canonical C-terminal secretion signal [10]. Deletion of SidJ’s internal secretion signal impairs *L. pneumophila* intracellular replication to the same extent as a loss-of-function mutation in *sidJ* [10]. The importance of temporal regulation of the SidE family of effectors by SidJ within host cells demonstrates the critical role of metaeffectors in the establishment of *L. pneumophila*’s intracellular replicative niche.

The metaeffector MesI (Lpg2505) was identified following high-throughput forward genetic screening for effector virulence phenotypes using transposon insertion sequencing (INSeq) [18]. *L. pneumophila* defective in *mesI* have a severe intracellular growth defect in both a natural amoebal host and mouse models of infection [18]. However, the virulence defect-associated absence of *mesI* is due solely to the activity of its cognate effector, SidI, since loss-of-function mutation in *sidI* rescues the growth defect of the ∆*mesI* mutant [18]. SidI is a cytotoxic effector that inhibits eukaryotic protein translation in vitro and contributes to activation of the heat shock response in *L. pneumophila*-infected cells [19]. We recently discovered that SidI possesses GDP-mannose-dependent glycosyl hydrolase activity and likely functions as a mannosyltransferase [20]. MesI is sufficient to abrogate both SidI-mediated toxicity and protein translation inhibition [18,20]. MesI binds SidI with nanomolar affinity and the interaction is characterized by a long half-life. MesI binds SidI on both N- and C-termini and does not impair interaction between SidI and its established binding partner, eEF1A (Figure 1) [20,21]. Despite almost complete abrogation of SidI-mediated translation inhibition, MesI only mildly attenuates SidI glycosyl hydrolase activity, suggesting that MesI does not function to inhibit SidI activity [20]. Although the regions of MesI important for binding the termini of SidI have yet to be defined, the crystal structure of MesI revealed a tetratricopeptide repeat (TPR) segment in MesI’s 6/7, 8/9, and 10/11 alpha-helices that form grooves predicted to play a role in SidI binding [21]. Whether the terminal regions of SidI bind to MesI through a large unilateral interface, or if multiple separate interaction sites exist on MesI is unknown. Whether MesI participates in SidI-mediated activation of the heat shock response is also unknown (Figure 1).

Urbanus and colleagues recently executed the most comprehensive effector toxicity suppression screen to date, resulting in the discovery of 17 effector-suppression pairs, including nine putative metaeffectors [22]. The researchers used a high-throughput yeast toxicity assay to screen over 108,000 pairwise effector-effector genetic interactions [22]. This study revealed the plasticity of metaeffector activity. In some cases, metaeffectors directly inactivate their cognate effector. For example, LegL1 deactivates its cognate effector through steric hindrance of its active site. Other metaeffectors, such as LupA and LubX, enzymatically modify their cognate effectors LegC3 and SidH (see above), respectively (Figure 1) [22]. LupA is a eukaryotic-like ubiquitin protease that catalyzes removal ubiquitin from LegC3 [22]. LegC3 is one of three *L. pneumophila* effectors that mimic eukaryotic Q-SNAREs to recruit vesicles coated with VAMP4 to the LCV [23,24]. How ubiquitiniylation influences LegC3 activity and the contribution of its regulation by its metaeffector LupA are both unknown.

This screen not only identified novel metaeffector pairs, but also unveiled diversity in effector function and regulation. SidP is a phosphatidylinositol-3-phosphate (PI3P) phosphatase [25]. However, SidP’s PI3P phosphatase activity is dispensable for binding and suppressing the toxicity of its cognate effector MavQ. The phosphatase activity of SidP resides within its N-terminal domain, and the C-terminal domain alone is sufficient for binding and regulation of MavQ. MavQ is a predicted phosphoinositide (PI) kinase, and together with SidP, likely regulates PI metabolism within host cells. Interestingly, SidP is toxic to yeast when expressed together with the effector Lem14; however, the role of Lem14 in SidP metaeffector activity and PIP metabolism has not been fully elucidated [22]. The putative role of MavQ as a PIP kinase, and the synergistic effects of SidP and Lem14 reveal a complex picture of effector regulation of host PIPs (Figure 1) [22].

LegA11 is a metaeffector of unknown function that binds and suppresses the toxicity of SidL. The N-terminal region of LegA11 contains ankyrin-repeats (PDB:4ZHB), which are canonically involved in protein-protein interactions [26,27]. Like SidI, SidL inhibits eukaryotic protein translation; however, SidL also inhibits actin polymerization when ectopically expressed in eukaryotic cells [28,29]. Aberrant organization of the actin cytoskeleton attenuates protein translation [30], but whether SidL-mediated translation inhibition is a consequence of impaired actin polymerization is unknown (Figure 1) [29,31]. The role of LegA11 in regulation of SidL function is unknown. Elucidating the mechanism by which LegA11 regulates SidL will likely shed light on SidL’s function and the importance of its spatiotemporal regulation.

The effector deamidases MavC and MvcA are both regulated by a single metaeffector, Lpg2149 (Figure 1). MavC and MvcA are functional antagonists that temporally regulate the activity of the host E2 enzyme, Ube2N. MavC catalyzes E1-independent monoubiquitination and inhibition of Ube2N [32]. However, prolonged inhibition of Ube2N is detrimental to *L. pneumophila* and is reversed through MvcA deubiquitination (Figure 1) [33]. Lpg2149 binds and inhibits the deamidase activity of both MavC and MvcA; however, the biological significance of this inhibition and influence on temporal regulation of Ube2N ubiquitination are unknown. Further investigation is required to uncover the role of Lpg2149 in *L. pneumophila* virulence. Collectively, these studies underlie the importance of metaeffectors in spatiotemporal regulation of *L. pneumophila* effector function.

## 3. What Makes a Metaeffector?

Classification of an effector as a metaeffector is based on two criteria, (1) binding; and (2) regulation of a cognate effector(s). Several metaeffectors, including LubX and SidJ, co-opt host proteins to regulate their cognate effectors. Moreover, LubX does not exclusively catalyze ubiquitination of SidH (see above), demonstrating the functional versatility of metaeffectors. Other metaeffectors, such as MesI, are able to regulate their cognate effectors in the absence of host components, but this does not preclude the involvement of host factors. A defining feature of metaeffectors is direct interaction with cognate effector proteins. However, other characteristics are shared amongst effector–metaeffector pairs.

### 3.1. Structure

In general, metaeffectors are smaller than their cognate effectors. This is a trend and not a rule, as several metaeffectors such as LegL1, LupA, and SidP are comparable in size to their targets (Table 1). It is also not uncommon for metaeffectors to contain interaction domains, such as the tetratricopeptide repeats (TPR) of MesI, ankyrin repeats of LegA11, or the leucine-rich repeats (LRR) of LegL1 [21,27]. These interaction domains are likely important for the interaction of effectors with their cognate effector. For example, the LRR of LegL1 forms a canonical horseshoe shape over RavJ’s active site, causing steric hindrance [22]. The ankyrin repeats in LegA11 likely facilitates protein–protein interactions (see above) [26,27]. Thus, several metaeffectors possess canonical protein–protein interaction motifs that are likely used to bind their cognate effector(s).

### 3.2. Proximity

Metaeffectors are typically encoded in close proximity to their cognate effector within the genome [22]. However, some exceptions exist, since *mavQ* is not encoded in the vicinity of either *sidP* or *lem14* [22]. Genomic analysis of 38 *Legionella* species revealed 143 effector pairs encoded in close proximity in at least two *Legionella* genomes. Nineteen of these effector pairs—including SidL-LegA11 and SidI-MesI—appear to have co-evolved; however, this number may be higher, as it only captures pairs found in multiple species and does not consider those unique to a single species [35]. Some effector pairs, such as SidL and LegA11, are always found in conjunction, while others, such as SidI and MesI, occasionally occur in solidarity [35]. *sidL* and *legA11* represent the most highly co-evolved effector pair in the *Legionella* genus [35]. Relatively little is known about transcriptional regulation of effector–metaeffector gene expression. Interestingly, *legA11* and *sidL* are encoded adjacent to each other, but on different strands of the chromosome and initiate in opposite directions. Elucidating the timing and quantity of effector and metaeffector gene expression can provide additional spatiotemporal insights into mechanisms of metaeffector-mediated regulation of effectors. While effector pairs are present across the *Legionella* genus [3], only *L. pneumophila* metaeffectors have been studied to date. Although all Legionella species studied to date replicate within an endoplasmic reticulum-derived LCV, whether species-specific differences affect metaeffector-effector regulation and function exist has yet to be elucidated.

## 4. Concluding Remarks

Although effectors are critical virulence factors for many Gram-negative bacterial pathogens, mechanisms by which effectors are regulated within host cells are poorly understood. Metaeffectors provide an additional layer of regulation and spatiotemporal fine-tuning of effector function. Although metaeffectors are currently unique to the *Legionella* genus, it is tempting to speculate that other pathogen virulence strategies involve metaeffectors. However, identification of metaeffectors is challenging, and relies on robust phenotypes resulting from effector dysregulation. Urbanus and colleagues conducted the most extensive effector-pair screen to date using a yeast expression model. However, other metaeffector–effector pairs may be incognito within this unnatural expression in the absence of a toxic effector phenotype [22]. Extreme functional redundancy within *L. pneumophila*’s effector repertoire creates challenges, as deletion of a single effector rarely leads to a discernable phenotype [6,18]. MesI and SidJ are two of less than a dozen effectors that are individually important for *L. pneumophila* intracellular replication. Thus, metaeffectors play a major role in the virulence strategy of *L. pneumophila,* which emphasizes the importance of both effector interplay and functional regulation. Metaeffectors represent a noncanonical effector regulatory system that is likely not unique to *L. pneumophila*. Identification of metaeffector and metaeffector-like functions has been contingent on observable phenotypes, such as toxicity or intracellular replication; however, scrutiny of genomic organization of effector genes may lead to identification of additional metaeffectors encoded by other *Legionella* species and other bacterial pathogens. Further investigation will undoubtedly reveal additional mechanisms of effector regulation arising from host-pathogen co-evolution, and could provide a foundation for development of anti-virulence therapeutics. 

## Figures and Tables

**Figure 1 pathogens-10-00108-f001:**
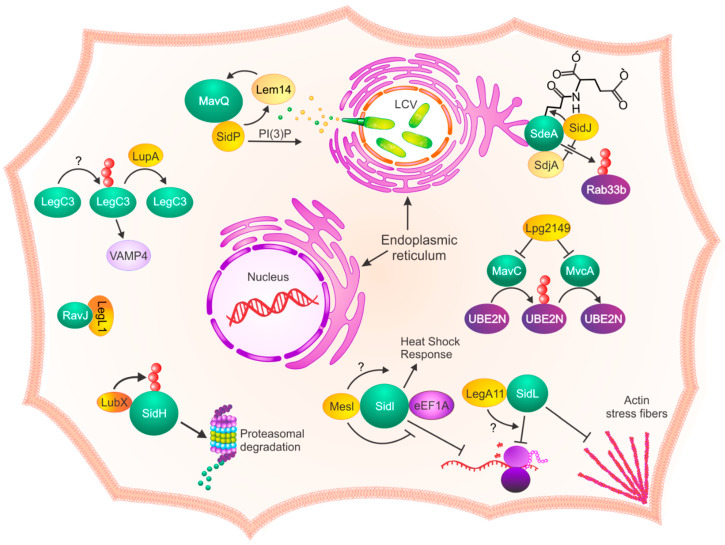
*L. pneumophila* metaeffectors exploit various modes of action against other effectors, many leading to the deactivation or degradation of their target. *L. pneumophila* which relies on complex regulation of effector synthesis and translocation to orchestrate successful host cell invasion, and it can be speculated that this intricacy applies to metaeffector regulation as well. Metaeffectors likely prevent overactivity of their target effectors, which can be detrimental to the *L. pneumophila* intracellular life cycle. While the activity of some effectors, such as the transglutamylation of SdeA by SidJ or interactions of SidI with MesI prevent toxicity attributed to their target, others like SidP and Lem14 with MavQ have a more complicated relationship with their target effector that has yet to be uncovered. Yellow, metaeffectors; Teal, effectors; Purple, host proteins and structures; Red, ubiquitin; Green-yellow, *L. pneumophila*.

**Table 1 pathogens-10-00108-t001:** Known *L. pneumophila* effector-metaeffector pairs and their activities.

Metaeffector	Gene ID	Activity	Size ^a^	Effector	Gene ID	Activity ^b^	Size ^a^	Refs.
LegA11/AnkJ	Lpg0436	Unknown	269	SidL/Ceg14	Lpg0437	Translation inhibitor	666	[23]
LegL1	Lpg0945	Competitive inhibition	296	RavJ	Lpg0944	Putative transglutaminase	391	[23]
Lem14	Lpg1851	Synergistic with SidP	220	MavQ	Lpg2975	Putative kinase	871	[23]
Lpg2149	Lpg2149	Unknown	119	MavC	Lpg2147	Ubiquitin-ase	482	[34]
				MvcA	Lpg2148	Deubiquitinase	426	
LubX	Lpg2830	E3 Ubiquitin Ligase	246	SidH	Lpg2829	SdhA homolog	2225	[4]
LupA	Lpg1148	Deubiquitinase	503	LegC3	Lpg1701	Glutamine (Q)-SNARE-like protein	506	[23,25]
MavE	Lpg2344	Unknown	208	YlfA/LegC7	Lpg2298	SNARE-like Protein	425	[23]
MesI	Lpg2505	Unknown	295	SidI/Ceg32	Lpg2504	Putative mannosyltransferase	942	[19,21,22]
SdbC	Lpg2391	Putative Lipase	434	SdbB	Lpg2482	Putative Lipase	448	[23]
SidJ	Lpg2155	Calmodulin-dependent transglutamylase	873	SidE	Lpg0234	Ubiquitin Ligases	1575	[9,10,11,12,15,16,17,18,23]
				SdeA	Lpg2157		1506	
				SdeB	Lpg2156		1926	
				SdeC	Lpg2153		1533	
SidP	Lpg0130	PI3P Phosphatase	822	MavQ	Lpg2975	Putative PIP Kinase	871	[23]

^a^ Protein size shown as number of amino acid residues; ^b^ Predicted activity determined using HHPred [34].

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
