# Peer review of "Affecting the Effectors: Regulation of Legionella pneumophila Effector Function by Metaeffectors"

_pathogens, 2021, doi:10.3390/pathogens10020108_

Round 1

Reviewer 1 Report

Dear Authors and Editors,

The manuscript written by Joseph and Shames is well organized and deals with a very interesting topic regarding Legionella pathogenesis. The authors skillfully reviewed the current knowledge on Legionella metaeffectors in a very concise way and discussed the main challenges in deciphering their contribution to virulence. Furthermore, this review provides a present-day bibliography on the subject. Therefore, I believe this manuscript is suitable for publication in Pathogens. I just have a minor comment for the authors: keywords are missing (line 17), please include some.

Author Response

Reviewer 1:

The manuscript written by Joseph and Shames is well organized and deals with a very interesting topic regarding Legionella pathogenesis. The authors skillfully reviewed the current knowledge on Legionella metaeffectors in a very concise way and discussed the main challenges in deciphering their contribution to virulence. Furthermore, this review provides a present-day bibliography on the subject. Therefore, I believe this manuscript is suitable for publication in Pathogens. I just have a minor comment for the authors:

We thank the reviewer for their feedback and positive feedback on our manuscript.

  1. keywords are missing (line 17), please include some.

We have added the keywords “Legionella, metaeffector, effector”

Reviewer 2 Report

The review entitled  “Affecting the effectors: Regulation of Legionella pneumophila effector function by metaeffectors “ is a concise study that report  the metaeffector function of Legionella, challenges associated with their identification and potential avenues to reveal the contribution of metaeffectors to bacterial pathogenesis.

In general the study is interesting and well written. However, there are some unclear things . I would like authors to address  (see a point by point discussion below).

In general:

correct Legionella with Legionella spp (Line 127)

The conclusions are intuitive and the findings don't support the aims of study.  Please, insert some comments to support the aims of review.

Author Response

Reviewer 2:

The review entitled  “Affecting the effectors: Regulation of Legionella pneumophila effector function by metaeffectors “ is a concise study that report  the metaeffector function of Legionella, challenges associated with their identification and potential avenues to reveal the contribution of metaeffectors to bacterial pathogenesis.

In general the study is interesting and well written. However, there are some unclear things . I would like authors to address  (see a point by point discussion below).

We thank the reviewer for their comments and have addressed them below. We believe the manuscript is improved.

In general:

  1. correct Legionellawith Legionella spp (Line 127)

We have edited this to read L. pneumophila.

  1. The conclusions are intuitive and the findings don't support the aims of study.  Please, insert some comments to support the aims of review.

We thank the reviewer for their suggestion and have updated the introduction better describe the aims of the review (lines 35-38)

“Here, we discuss current knowledge pertaining to L. pneumophila metaeffectors and conclude with the importance of future investigation into these important virulence factors both within the Legionella genus and other bacterial pathogens.”